# Feasibility, Uptake, and Results of COVID-19 Antigen Rapid Diagnostic Tests among Refugees and Migrants in a Pilot Project in North-West Syria

**DOI:** 10.3390/tropicalmed8050281

**Published:** 2023-05-16

**Authors:** Hassan Ghawji, Mohamad Nihad AlYousfi, Srinath Satyanarayana, Nevin Wilson, Laila Tomeh, Hussam Alkhellov, Sali Hasan, Sanjay Sarin, Kekeletso Kao

**Affiliations:** 1International Organization of Migration, Suheil Majdoubeh 12, Tila’a Al-Ali, Amman P.O. Box 4880, Jordan; malyousfi@iom.int (M.N.A.); drsrinaths@gmail.com (S.S.); nwilson@iom.int (N.W.); 2International Organization of Migration, Sokak No:15 Tugay Şehitkamil, Güvenevler Mahallesi, Gaziantep 29069, Turkey; ltomeh@iom.int; 3Hand in Hand for Aid and Development, Sokak No:10 27070 Şahinbey/Gaziantep, Binevler, Gaziantep 81041, Turkey; h.khellow@hihfad.org (H.A.); s.hasan@hihfad.org (S.H.); 4Foundation for Innovative New Diagnostics FIND, Av De Budé 16, 1202 Geneva, Switzerland; sanjay.sarin@finddx.org (S.S.); kekeletso.kao@finddx.org (K.K.)

**Keywords:** COVID-19, rapid diagnostic tests, refugees, prevention, Ag-RDTs, PCR

## Abstract

North-west Syria (NWS) is a conflict-affected and unstable area. Due to its limited health infrastructure, accessing advanced COVID-19 testing services is challenging. COVID-19 antigen rapid diagnostic tests (Ag-RDTs) have the potential to overcome this barrier. A pilot project was implemented to introduce Ag-RDTs in NWS, aiming to determine the feasibility, uptake, and results of Ag-RDTs and identify facilitators and barriers to testing with Ag-RDTs. A cross-sectional study design involving secondary analysis of data collected during the project was employed. A local non-governmental organization implemented 25,000 Ag-RDTs that were conducted cross-border by trained community health workers. In total, 27,888 eligible individuals were enrolled, 24,956 (89.5%) consented to test, and 121 (0.5%) were COVID-19-positive. The highest positivity was observed among those with severe COVID-19 symptoms (12.7%), with respiratory illnesses (2.5%), enrolled at hospitals in Afrin (2.5%), and healthcare workers (1.9%). A non-random sample of 236 individuals underwent confirmatory RT-PCR testing. Observed sensitivity, specificity, and positive and negative predictive values were 80.0%, 96.1%, 91.4%, and 90.3%, respectively. Challenges included obtaining informed consent and conducting confirmatory testing. Ag-RDTs represent a feasible screening/diagnostic tool for COVID-19 infections in NWS, with nearly 90% uptake. Embedding Ag-RDTs into COVID-19 testing and screening strategies would be highly beneficial.

## 1. Introduction

The Middle East and North Africa (MENA) countries experienced high morbidity and mortality during the COVID-19 pandemic, with more than 18 million individuals infected and over 750,000 deaths [1]. The pandemic led to a considerable burden on the health systems of MENA countries that cater to refugees and migrants, leading to inadequate diagnostic testing. This has resulted in refugees and migrants experiencing disproportionately limited access to COVID-19 testing [2,3]. Therefore, it is likely that the number of people infected and deaths due to COVID-19 were higher than reported [4].

North-west Syria (NWS), with a population of 4.6 million, is one of the world’s most politically unstable geographical areas. Nearly 2.9 million people in this area comprise internally displaced populations (IDPs) due to war. This population has faced several challenges in accessing COVID-19 testing services due to deficiencies in the health infrastructure [5,6]. However, there is minimal scientific evidence about the barriers to COVID-19 testing in this setting [5]. Studies from neighboring countries have highlighted that displaced populations face financial burdens and fear of stigma, discrimination, and isolation while undergoing COVID-19 tests [6]. In addition, these studies indicate that the pandemic was mainly viewed by refugees as “overestimated”, and many hold conspiracy theory beliefs that have affected the uptake of COVID-19 testing in these populations [6,7].

Antigen rapid diagnostic tests (Ag-RDTs) are easy to perform, require minimal resources and infrastructure, can yield results in just 15 to 20 min, and are cheaper than molecular reverse transcription polymerase chain reaction (RT-PCR)-based tests [8]. Although they are more accurate than Ag-RDTs, RT-PCR tests require advanced resources and infrastructure [9]. Nevertheless, Ag-RDTs can have high diagnostic utility in tackling the spread of COVID-19, especially during the first seven days of symptom onset [10]. Thus, they can be effective in settings with limited access to RT-PCR testing.

In this context, the International Organization for Migration (IOM), in collaboration with the Foundation for Innovative New Diagnostics (FIND [11], implemented a pilot project for assessing the feasibility of using Ag-RDTs in NWS. This pilot project aimed to facilitate easy access to COVID-19 testing and to understand the barriers to and best practices for diagnostic testing in this displaced population setting. The pilot project was implemented in NWS in partnership with a local non-governmental organization—Hand in Hand for Aid and Development (HiHFAD)—and involved the distribution and use of 25,000 Ag-RDTs at health facilities, refugee camps, informal tented settlements, and in outreach settings.

As part of this pilot project, operational research was undertaken to document the experience of implementing Ag-RDTs in NWS. The main objectives were to (a) determine the feasibility of conducting Ag-RDTs within camps and health facilities; (b) determine the willingness of eligible participants to accept Ag-RDTs; (c) describe the results of the Ag-RDTs; (d) identify the enablers and barriers to the use of Ag-RDTs in these settings from the perspective of the healthcare workers who were involved in conducting these tests within NWS.

## 2. Materials and Methods

### 2.1. Study Design

This cross-sectional study involved secondary analysis of data collected as part of the pilot project’s recording, reporting, and monitoring processes.

### 2.2. Study Population Settings

As of 2023, the population of NWS is estimated to be around 4.6 million people [12]. The number of individuals needing humanitarian assistance in NWS increased from 3.4 to 4.1 million between 2021 and 2022. Women and children make up 80% of NWS’s population [13]. About 2.9 million people in NWS are IDPs, and the majority are food insecure, with more than 3 million of these individuals unable to meet their basic needs [12,13]. In addition, as of December 2022, the latest health updates in NWS suggest there are continuing large increases in daily COVID-19 cases, with a limited humanitarian response [14,15].

### 2.3. Ag-RDT Project Settings

The Ag-RDT project was implemented between September and November 2022 in various camps, mobile clinics, and health facilities in three districts within NWS: Afrin, Idleb, and Jarablus/Albab. The Ag-RDT used was the World Health Organization (WHO)-prequalified Sure Status COVID-19 Antigen Card Test^®^ (professional version), manufactured by Premier Medical Corporation (PMC) India, which has a ‘manufacturer reported’ sensitivity of 94.16% and specificity of 100%. An implementing partner organization, HiHFAD, was involved in distributing the Ag-RDTs within the study districts. HiHFAD deployed 48 community health workers (CHWs), comprising 50% males and 50% females, in 24 teams, with an additional 4 team leaders to oversee the implementation of 25,000 Ag-RDTs. All CHWs and team leaders underwent a two-day training program about how to perform the Ag-RDTs. A context-specific algorithm (shown in Figure 1), consistent with WHO recommendations for low-prevalence settings, was used to interpret the Ag-RDT results.

### 2.4. Eligibility Criteria

Individuals from the target communities who met the following criteria were offered an Ag-RDT: (a) patients presenting with COVID-19 symptoms to health facilities (primary healthcare centers (PHCs), hospitals, and tuberculosis (TB) centers) within seven days of onset of symptoms; (b) symptomatic or asymptomatic individuals who were contacts of a COVID-19-infected individual; (c) refugees and migrants in densely populated areas where the risk of COVID-19 transmission was greater; (d) healthcare workers; (e) any other individuals who voluntarily sought a COVID-19 test. All individuals who met the eligibility criteria were enrolled for Ag-RDT testing; tests were only conducted once a participant had provided written informed consent.

### 2.5. Data Recording and Reporting

A data recording and reporting system was designed to capture the demographic and clinical characteristics of participants undergoing testing. The data were initially recorded as hard copies (in logbooks). They were later entered into an online digital platform created using the KOBO Toolbox® (Harvard Humanitarian Initiative, Cambridge, MA, USA) to enable real-time monitoring of the project’s implementation.

### 2.6. Project Supervision

To supervise and monitor the project’s implementation activities, supervisory visits were made by the four project leaders, during which they interacted with CHWs and participants involved in performing or undergoing the Ag-RDTs. In addition, a supervisory visit by the project coordinator was made to the head offices of IOM and HiHFAD in Turkey, which included interviews with team leaders and CHWs to discuss the progress made and challenges faced. Detailed reports of field visits were prepared and used as the basis for determining whether any corrective actions were necessary. In addition, the project coordinators reviewed the data entered by the CHWs into the Kobo application on a weekly basis to assess any gaps in the implementation or the data entered and provide feedback to address any deficiencies.

### 2.7. Study Sample

All individuals in the study areas who were eligible for an Ag-RDT, regardless of their age, were included in the pilot project, which was implemented between September and November 2022. The pilot project enrolled 27,888 individuals who were eligible to be offered an Ag-RDT.

### 2.8. Data Variables

For the first three objectives of the study, the variables included the name of the health facility and participants’ date of enrollment, age, gender, presence or absence of symptoms, reason for testing, vaccination status, response to being asked to provide consent, Ag-RDT results, date of the test, presence or absence of risk factors, and confirmatory RT-PCR test result (if conducted) and its date. For the fourth objective, data gathered during the supervisory visits by the project coordinators were used to obtain information about the facilitators and barriers to the implementation and uptake of Ag-RDTs in the field.

### 2.9. Data Analysis and Interpretation

To describe the feasibility, uptake, results of Ag-RDTs, and demographic and clinical data of participants, these variables were summarized as frequencies and percentages using R and RStudio statistical software. Associations between demographic and clinical characteristics (age, gender, reason for testing, and district) and the uptake of tests were assessed using bivariable and multivariable logistic regression models. The model fit was assessed using the Hosmer–Lemeshow goodness of fit test. Inferences around feasibility were based on the ease with which the CHWs could conduct the tests correctly, as per the designed algorithm, in different health facilities and geographical areas. These assumptions were based on discussions with CHWs and were documented in the supervisory visits. Furthermore, inferences around the uptake of Ag-RDTs were assessed based on the number and proportion of eligible individuals who gave consent to undergo the tests. Finally, to identify the facilitators and barriers to testing with Ag-RDTs, we produced a narrative table of the lessons learned and challenges encountered by CHWs when implementing the project, as determined by the project coordinators.

### 2.10. Ethical Issues

This project was implemented after obtaining administrative approval from the Turkish directorates (Hatay and Gaziantep) responsible for implementing all health activities within NWS. Ethics approval was obtained from the Institutional Review Board committee of Rayak Hospital, Lebanon, with the reference code ECO-R-35. Confidentiality of all data obtained from the pilot project’s recording system has been maintained. Consent was obtained from all individuals prior to them taking a test and to use their data for operational research. No names or identifying information of any person have been used in the analysis. Only aggregate data were used to disseminate the results of the study to all stakeholders.

## 3. Results

Between September and November 2022, 27,888 individuals in the study sites were eligible to enroll in the project and were offered an Ag-RDT. The demographic and clinical characteristics of all enrolled participants and the COVID-19 positivity rates were analyzed by gender, age group (Table 1), reason for testing, vaccination status, duration, severity of symptoms (Table 2), and testing facility (Table 3).

Overall, 24,956 (89.5%) of the 27,888 individuals enrolled consented to undergo an Ag-RDT, with 121 (0.5%) testing positive for COVID-19 by Ag-RDT. The study population comprised more females (57.1%) than males (42.9%), and approximately 74% of participants were between the ages of 18 and 49 years. Consent to take a test was lower among males (86.7%) than females (90.3%).

The main reasons for testing at the enrollment stage were being a refugee living within a densely populated and crowded area (44.5%), followed by having COVID-19 symptoms (43.1%). The highest proportions of positive cases in individuals who consented to test were observed among those with severe COVID-19 symptoms (12.7%), those with respiratory illnesses (2.5%), and healthcare workers (1.9%).

Overall, 12,026 (43.1%) individuals reporting having COVID-19 symptoms, of whom 11,777 (97.9%) reported mild symptoms, while 98.9% of participants reported the onset of symptoms within the previous seven days or less. Testing the 136 (1.1%) participants who had experienced symptoms for more than 7 days was performed because of challenges in convincing them they were excluded from the study simply because of their long duration of symptoms. There were 17,166 (61.6%) participants who had not received a single dose of any COVID-19 vaccine.

The majority (86.0%) of enrollments occurred at health facilities. The positivity rate among those who consented to testing was highest among participants from hospitals in Afrin (2.5%).

The demographic and clinical characteristics associated with the uptake of Ag-RDTs are shown in Table 4. Our bivariate analysis demonstrated that males were less likely than females to consent to an Ag-RDT. Participants aged 5 to 17 years were 1.35 times more likely to consent to having an Ag-RDT than those aged 18 to 34 years. Most participants who had a reason for testing, such as COVID-19-like symptoms, contact with a confirmed case of COVID-19, or recent travel, were more likely to agree to have an Ag-RDT than those without a reason for testing. However, participants’ willingness to have an Ag-RDT was considerably lower for IDPs than non-IDPs in NWS. Participants with severe COVID-19-like symptoms were 1.6 times more likely to agree to have an Ag-RDT than those with mild symptoms. Consent rates were highest at facilities in Jarablus, with participants at health facilities and PHCs in Afrin and Idlib being less likely to consent to having an Ag-RDT.

In the multivariable analysis, gender remained a significant predictor of consenting to having an Ag-RDT, after considering potential confounders (odds ratio (OR) = 0.69, 95% confidence interval (CI) (0.645, 0.753)). After controlling for confounders, we observed that being younger (less than 18 years old) was significantly associated with being more likely to consent to having an Ag-RDT when compared with individuals aged 18 to 34 years. Health workers, refugees, and symptomatic individuals were more likely to agree to having an Ag-RDT than other groups. A significant predictor of uptake was the severity of symptoms, as individuals with severe symptoms were 2.3 times more likely to consent to having an Ag-RDT than people with mild symptoms.

Of those who underwent an Ag-RDT, a non-random sample of 239 individuals also had a confirmatory RT-PCR test (Table 5). This sample was selected based on the availability and accessibility of RT-PCR testing services for participants during the implementation period. This included 72/121 (59.5%) individuals who received a positive Ag-RDT result and 166/24,811 (6.7%) individuals who received a negative Ag-RDT result.

Of the participants who underwent a confirmatory RT-PCR test (N = 239), the project team could not collect/receive information about the RT-PCR test results for three individuals. Therefore, after excluding these individuals, 236 individuals had results for both Ag-RDT and RT-PCR tests (Table 5). Based on this dataset, the ‘observed’ sensitivity of Ag-RDTs was 80.0% (95% CI: 71.2–88.7%) (64/80); specificity was 96.1% (95% CI 93–99%) (149/155); the positive predictive value was 91.4% (95% CI 84.8–97.9%) (64/70); and the negative predictive value was 90.3% (95%CI 85.7–94.8%) (149/165). The overall flow of participants under the project is given in Figure 2.

The reported barriers and facilitators encountered during the implementation of the project are shown in Table 6 and Table 7. The key challenges were obtaining informed consent, referral for confirmatory RT-PCR testing, and recording and reporting the data for the project.

## 4. Discussion

This is one of the first studies from NWS describing the use of COVID-19 Ag-RDTs in a pilot project. Individuals who were eligible for testing were contacted, and the project’s target quota of participants was achieved within a very short, 2-month implementation period. The wide variations seen in the numbers enrolled at the nine study sites are due to the differences in the population distribution in these geographical areas. Enrollment at PHCs and hospitals was higher than at camps/mobile clinics, as all of the health facilities and hospitals in NWS are distributed inside the camps. As reported by the implementing partner, only a few camps lack these facilities. All nine facilities selected successfully deployed all their Ag-RDTs, indicating good feasibility of using these tests at all sites within NWS, distributed through CHWs. Furthermore, the interviews with CHWs during the supervisory visits did not reveal any major challenges in deploying or performing these tests, adding to the global body of evidence for the feasibility of rapid deployment of Ag-RDTs to inform essential infection control and prevention measures in all settings [16,17].

Almost 90% of participants were willing to take an Ag-RDT. This was mainly due to the positive communication from the CHWs about the advantages of Ag-RDTs and the need to ensure there was no break in COVID-19 surveillance. This communication helped to engender positive perceptions about the Ag-RDTs and facilitated their uptake. The uptake of Ag-RDTs was higher among individuals with COVID-19 symptoms, healthcare workers, refugees, individuals who had received two or more doses of vaccine, and individuals identified at TB clinics than it was among other groups. Previous studies have shown good overall uptake of Ag-RDTs among different populations, especially among symptomatic individuals [18,19]. The reasons for the lower uptake of Ag-RDTs among those aged less than 18 years and among individuals in Afrin and Idleb districts could not be determined from the interviews with CHWs. This is an area that will therefore need further investigation through the use of qualitative research methods.

The overall positivity rate within our sample was very low (just 0.5%), indicating low levels of COVID-19 prevalence at the community level. The highest positivity was observed among individuals with severe COVID-19 symptoms (12.7%), followed by individuals with respiratory illnesses (2.5%), participants enrolled at hospitals in Afrin (2.5%), and healthcare workers (1.9%). On one hand, these findings indicate the high-risk groups who should be targeted for COVID-19 testing in these low-prevalence/-transmission settings. On the other hand, these findings also suggest that some geographical regions or pockets of persisting COVID-19 infection could lead to a flare-up of COVID-19 transmission if there were to be a new variant of the virus. Therefore, continuous surveillance is needed. Data about positivity rates and the association with participants’ various characteristics can play an essential role in surveillance activities, especially for the screening and diagnosis of COVID-19 infection in these high-risk groups.

Unfortunately, a formal assessment of the sensitivity and specificity of the selected Ag-RDTs when deployed in the field could not be carried out due to challenges in identifying an academic partner in this setting. Nevertheless, the observed sensitivity and specificity of the Ag-RDTs in relation to the RT-PCR appears to be in line with those reported for other studies involving the same Ag-RDTs as used in our pilot project [20,21]. It should be noted, however, that the performance of Ag-RDTs varies among brands [22]. Therefore, policymakers should pay careful attention when selecting suitable Ag-RDTs for testing programs. The sensitivity and negative predictive values observed in this pilot project, which were lower than the manufacturer-reported values, could be due to the quality of the nasopharyngeal samples collected, and the difficulties processing the collected samples and the reagents in the test kits [23,24,25,26,27].

Such deficiencies must be identified and rectified through training and supportive supervision in any future projects. However, considering the high specificity (96.1%) and good negative predictive value (90.3%), screening for COVID-19 using Ag-RDTs results would be a reasonable approach in this setting, whether or not a confirmatory PCR test is used.

It was only possible to perform confirmatory RT-PCR tests (as per the algorithm) in 60% of individuals who had a positive Ag-RDT result. This indicates operational challenges in following the complete algorithm needed for the optimal deployment of Ag-RDTs. Similar observations were made in other studies that evaluated the implementation of Ag-RDTs [28,29]. Therefore, there is a need to enhance both the capacity and accessibility of RT-PCR testing within the health system catering to these populations. If this is not completed, clinical and public health decisions in relation to COVID-19 infection/transmission will have to be based on Ag-RDTs alone.

Finally, 61.6% of our sample reported they had not received a single dose of COVID-19 vaccine. WHO reported that the proportion of people in NWS who had not received a single dose of COVID-19 vaccine during our implementation period was 86.4% [30], showing that there are considerable global inequities in vaccination coverage in these populations.

### Strengths and Limitations

The major strength of this study is that it was based on data from a pilot project implemented under routine programmatic conditions and using the existing health service delivery mechanisms. Therefore, the study’s findings in relation to feasibility, uptake, Ag-RDT performance, and positivity rates likely reflect ground-level realities. The study’s major limitations are as follows: First, the project was conducted in a low-prevalence/-transmission setting. Therefore, the results may not be replicable if there is a surge in COVID-19 prevalence. Second, the implementation challenges were identified, and facilitators’ data were collected during routine supervision visits. As a result, the information provided by CHWs may have been biased if they felt they had failed in any way or were uncomfortable reporting any critical issues to their supervisors. Third, due to challenges/restrictions in meeting the members of the community, their perspectives about the feasibility and acceptability of Ag-RDTs could not be fully captured. This is an area for future research.

## 5. Conclusions

This pilot project conducted in NWS demonstrated that using Ag-RDTs and delivering them through CHWs as a screening/diagnostic tool for COVID-19 infections is feasible. The data for uptake and positivity rates indicate that Ag-RDTs can be targeted to health facilities; individuals with respiratory, febrile, or COVID-19 symptoms; contacts of COVID-19 cases; and health workers. The project also highlighted the need for a comprehensive strategy to independently use Ag-RDT results to screen for COVID-19 infections among refugees and in IDP settings. Finally, the study confirms the need for continuous surveillance and enhancement of COVID-19 vaccination coverage.

## Figures and Tables

**Figure 1 tropicalmed-08-00281-f001:**
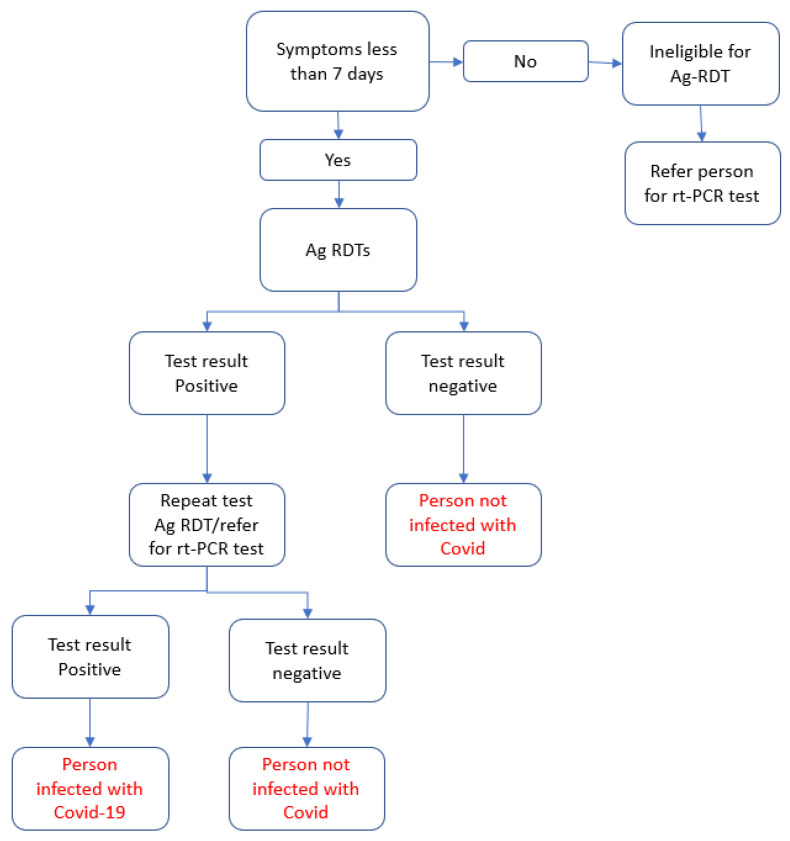
Algorithm used to interpret antigen rapid diagnostic test (Ag-RDT) results in a pilot project conducted in north-west Syria (NWS).

**Figure 2 tropicalmed-08-00281-f002:**
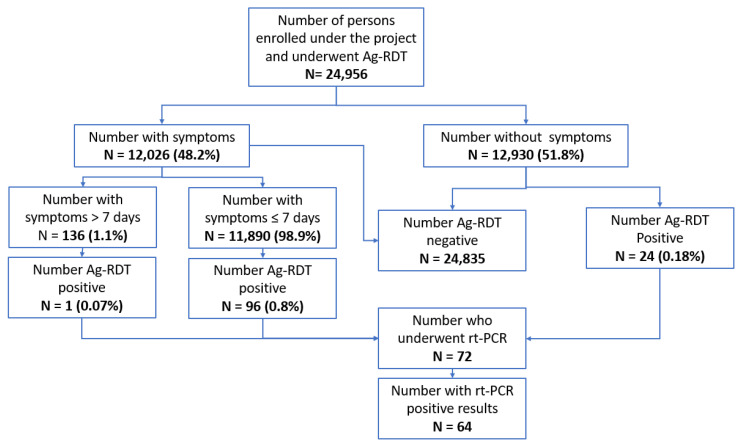
Antigen rapid diagnostic test (Ag-RDT) results in a pilot project conducted in north-west Syria (NWS).

**Table 1 tropicalmed-08-00281-t001:** Demographic and clinical characteristics of enrolled individuals and their COVID-19 positivity rates, by gender and age group.

	Participants		Consented to an Ag-RDT		COVID-19-Positive Results among Those Who Underwent an Ag-RDT	
Demographic and Clinical Characteristics	N	(%)	N	(%)	N	(%)
**Total**	**27,888**	**100.0%**	**24,956**	**89.5%**	**121**	**0.5%**
**Gender**						
Female	15,913	57.1%	14,474	90.3%	60	0.4%
Male	11,959	42.9%	10,467	86.7%	61	0.6%
Other	16	0.1%	15	93.8%	0	0.0%
**Age (years)**						
<5	685	2.5%	557	81.3%	6	1.1%
5–17	2217	7.9%	2038	91.9%	21	1.0%
18–34	13,569	48.7%	12,087	89.1%	44	0.4%
35–49	7002	25.1%	6311	90.1%	34	0.5%
50–64	3338	12.0%	2999	89.8%	12	0.4%
>64	1077	3.9%	964	89.5%	4	0.4%

**Table 2 tropicalmed-08-00281-t002:** Demographic and clinical characteristics of enrolled individuals and their COVID-19 positivity rates by reason for testing, vaccination status, and duration and severity of symptoms.

	Participants		Consented to an Ag-RDT		COVID-19-Positive Results among Those Who Underwent an Ag-RDT	
**Demographic and Clinical Characteristics**	**N**	**(%)**	**N**	**(%)**	**N**	**(%)**
**Total**	**27,888**	**100.0%**	**24,956**	**89.5%**	**121**	**0.5%**
**Reason for Testing ***						
Febrile illness	307	1.1%	288	93.8%	2	0.7%
Respiratory illness	198	0.7%	177	89.4%	4	2.3%
Other (seeking assurance)	1095	3.9%	1087	99.3%	0	0.0%
Healthcare worker	731	2.6%	671	91.8%	13	1.9%
Is a refugee	12,419	44.5%	10,811	87.1%	10	0.1%
Is a migrant	29	0.1%	28	96.6%	0	0.0%
Has COVID-19 symptoms	12,026	43.1%	11,054	91.9%	97	0.9%
Contact of a COVID-19-infected person	1672	6.0%	1393	83.3%	10	0.7%
Recent travel	93	0.3%	90	96.8%	1	1.1%
Identified at a TB clinic	123	0.4%	119	96.7%	0	0.0%
Admitted to ICU	307	1.1%	21	100.0%	0	0.0%
**Vaccination Status**						
No doses	17,166	61.6%	15,379	89.6%	79	0.5%
One dose	4802	17.2%	4246	88.4%	15	0.4%
Two doses	4504	16.2%	4049	89.9%	18	0.4%
More than two doses	1416	5.1%	1282	90.5%	9	0.7%
**Duration of Symptoms**	**N = 12,026 ****	**(%)**	**N = 11,054 ^$^**	**(%)**		
≤7 days	11,890	98.9%	10,928	91.9%	96	0.9%
>7 days	136	1.1%	126	92.6%	1	0.8%
**Severity of Symptoms**	**N = 12,026 ****	**(%)**	**N = 11,054 ^$^**	**(%)**		
Mild	11,777	97.9%	10,818	91.9%	67	0.6%

ICU, intensive care unit; TB, tuberculosis; * Multiple reasons were possible; therefore, the total may add up to >100%. ** N here includes only enrolled participants with COVID-19 symptoms. ^$^: N here includes symptomatic participants who consented to undergo an Ag-RDT.

**Table 3 tropicalmed-08-00281-t003:** Demographic and clinical characteristics of enrolled individuals and their COVID-19 positivity rates, by testing facility.

	Participants		Uptake of Ag-RDT		COVID-19-Positive Results among Those Who Underwent an Ag-RDT	
Demographic and Clinical Characteristics	N	(%)	N	(%)	N	(%)
**Total**	**27,888**	**100.0%**	**24,956**	**89.5%**	**121**	**0.5%**
**Testing facility**						
Idleb—PHC	8732	31.3%	7435	85.1%	24	0.3%
Idleb—hospital	8152	29.2%	6998	85.8%	38	0.5%
Idleb–camps/mobile clinic	20	0.1%	16	80.0%	0	0.0%
Afrin—PHC	3845	13.8%	3438	89.4%	37	1.1%
Afrin—hospital	278	1.0%	228	82.0%	5	2.5%
Afrin—camp/mobile clinic	441	1.6%	433	98.2%	0	0.0%
Jarablus/Albab—PHC	3112	11.2%	3105	99.8%	7	0.2%
Jarablus/Albab—hospital	2274	8.2%	2269	99.8%	9	0.4%
Jarablus/Albab—camp/mobile clinic	1034	3.7%	1034	100.0%	1	0.1%

ICU, intensive care unit; PHC, primary health center; TB, tuberculosis.

**Table 4 tropicalmed-08-00281-t004:** Demographic characteristics associated with uptake of Ag-RDTs in NWS.

Demographic and Clinical Characteristics	Bivariable Analysis		Multivariable Analysis *		
	OR	95% CI	Adj. OR	95% CI	*p*-Value
**Gender**					
Female	Reference		Reference		
Male	0.69	(0.645, 0.753)	0.78	(0.72, 0.84)	<0.001
Others	1.49	(0.3, 26.9)	0.97	(0.11, 8.18)	0.983
**Age group (years)**					
<5	0.53	(0.43, 0.65)	0.62	(0.50, 0.77)	<0.001
5–17	1.39	(1.19, 1.64)	1.7	(1.43, 2.01)	<0.011
18–34	Ref		Ref		
35–49	1.11	(1.01, 1.23)	1.06	(0.96, 1.17)	0.209
50–64	1.08	(0.95, 1.23)	1.04	(0.91–1.18)	0.524
>64	1.04	(0.85, 1.28)	1.14	(0.42, 1.40)	0.204
**Reason for testing**					
Has symptoms	1.6	(1.479, 1.738)	2.63	(1.94, 3.55)	<0.001
Contact of a COVID-19 case	1.6	(1.479, 1.738)	1.4	(1.00, 1.95)	0.045
Healthcare worker	1.32	(1.01, 1.72)	2.86	(1.94, 4.22)	<0.001
Member of an IDP	0.63	(0.58, 0.68)	2.32	(1.70, 3.15)	<0.001
Is a migrant	3.29	(0.44, 24.2)	2.92	(0.35, 23.9)	0.317
Identified at a TB clinic	3.5	(1.29, 9.5)	12.27	(4.32, 34.8)	<0.001
Febrile illness	1.79	(1.12, 2.85)	1.41	(0.78, 2.54)	0.243
Respiratory illness	0.99	(0.63, 1.55)	1.55	(0.90, 2.65)	0.107
Recent travel	3.53	(1.12, 11.17)	1.46	(0.41, 5.23)	0.559
**Vaccination status**					
No doses	Ref				
One dose	0.88	(0.8, 0.98)	1.03	(0.93, 1.14)	0.564
Two doses	1.03	(0.92, 1.15)	1.15	(1.02, 1.29)	0.016
More than two doses	1.11	(0.93, 1.34)	1.3	(1.07, 1.58)	0.006
Testing district					
Idleb	0.011	(0.006, 0.019)	0.008	(0.004, 0.016)	<0.001
Afrin	0.017	(0.009, 0.029)	0.015	(0.008, 0.027)	<0.001
Jarablus	Ref		Ref		

Adj. OR, adjusted odds ratio; CI, confidence interval; IDP, internally displaced population; OR, odds ratio; TB, tuberculosis; * The *p* value of the Hosmer–Lemeshow goodness of fit test for the multivariable logistic regression model was <0.001, indicating that the odds ratios and adjusted odds ratios do not provide a good estimate of the association between the demographic characteristics. This means there could be several unmeasured variables that affect the uptake of Ag-RDTs, and, therefore, the magnitude of the odds ratios and adjusted odds ratios in Table 4 should be interpreted with caution.

**Table 5 tropicalmed-08-00281-t005:** Comparison of Ag-RDT and RT-PCR results in individuals who had both tests.

	RT-PCR Result			
Ag-RDT Result	Positive	Negative	Not Available	Total
Positive	64	6	2	72
Negative	16	149	1	166
Not available	0	1	0	1
Total	80	156	3	239

**Table 6 tropicalmed-08-00281-t006:** Challenges encountered during the Ag-RDT pilot project in NWS.

Domain	Challenge
Identification of eligible participants.	Excluding participants who had experienced symptoms for more than 7 days.
Obtaining informed consent.	The most common reason for refusing a test was fear of pain and discomfort based on a previous unpleasant experience when taking a test, followed by the belief that the COVID-19 pandemic was over. Other possible reasons included conspiracy theories, fear of isolation, and stigma.
Conducting the test.	No major challenges were detected.
Acceptability of positive/negative results.	No major challenges were detected.
Referral for RT-PCR testing.	Only a small number of RT-PCR tests were performed in collaboration with the Assistance Coordination Unit (ACU), the representing local NGO for COVID-19 RT-PCRs, as policies in NWS had restricted the overall use of COVID-19 PCR tests due to the low prevalence of COVID-19 at that time.
Supply chain management issues with Ag-RDTs or ancillary items.	No major challenges were detected.
Recording and reporting.	This was carried out using mobile phones, which was challenging, especially during the data cleaning stage. The use of laptops was a more convenient choice.
Mobility.	Initially, it was challenging for CHWs to move to implementing sites. However, this challenge was mitigated by the implementing partner by scheduling rides with their fleet of cars.

CHW, community health worker; NGO, non-governmental organization.

**Table 7 tropicalmed-08-00281-t007:** Facilitators that contributed to the good feasibility and uptake of Ag-RDTs.

Domain	Facilitators
Overall positive perceptions from the NWS community in relation to Ag-RDTs when compared with PCR and other tests.	Ag-RDTs yield results in a few minutes, giving participants rapid assurance about their health status. Ag-RDTs are performed on-site, so there is no need to travel to a laboratory to take a test.
Engagement of CHWs with previous experience of the COVID-19 response.	The implementing partner engaged CHWs who had previous experience of the COVID-19 response in NWS, which eased the implementation process for the Ag-RDT project.
Convenience in performing confirmatory PCRs when possible.	The implementing partner, HiHFAD, carried out some free-of-charge confirmatory RT-PCR tests in collaboration with a local NGO in NWS. Ag-RDT and RT-PCR samples were collected on the spot from each participant, in the presence of workers from HiHFAD and ACU, mitigating the financial and mobility challenges IDPs in NWS face when undergoing laboratory tests.
The common culture between health workers and participants	A shared culture between health workers and participants made it easier for participants to trust and accept the Ag-RDT implementation process and the test results.
Increasing the motivation of participants to take an Ag-RDT.	This was achieved through on-the-spot education and awareness sessions during the implementation of the project

CHW, community health worker; NGO, non-governmental organization.

## Data Availability

The clean set of raw data presented in this study are available from the corresponding author in MS Excel format upon reasonable request. The data are not publicly available due to data protection and privacy policies following the UN Migration Agency (IOM) standards.

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
