# Peer review of "Feasibility, Uptake, and Results of COVID-19 Antigen Rapid Diagnostic Tests among Refugees and Migrants in a Pilot Project in North-West Syria"

_tropicalmed, 2023, doi:10.3390/tropicalmed8050281_

Round 1
Reviewer 1 Report
Overall, an interesting and easy to read work. However, some points are unclear and should be improved or supplemented: Which Ag-RDT was used and what diagnostic properties are reported for these? Which PCR was used and what are its diagnostic properties? According to which exact criteria were the patients selected for the PCR tests? Why is "refugee" such a significant reason for a COVID test. How many of these people had COVID symptoms? Please insert a graph analogous to Figure 1 in which the respective test results are noted. Add a description of the model for the multivariate analysis, including the goodness of fit properties of the model. Based on the known diagnostic properties of AG-RDT and PCR, calculate positive and negative predictive values.
Reviewer 2 Report
The authors assessed if SARS-CoV-2 Ag-RDTs were capable of implementing in Northwest Syria. The study might be useful for the colleagues in similar local settings and countries shared similar background. I have some comments/queries so that the quality of the manuscript should be improved.
Major comments:
- Line 62: the authors should define the term ‘high diagnostic utility’, i.e. require less resources and infrastructure? in addition, the authors can elaborate more for the advantages of using Ag-RDTs, i.e. easy to use, short turn-around time, results can be interpreted without using equipment between 15-30 minutes, cheap, etc.
- line 80: ‘describe the uptake of Ag-RDTs from eligible persons’, I am not sure if I fully understand the term uptake of Ag-RDTs. After going through the manuscript, I try the rephrase it for you. ‘assess the willingness to perform Ag-RDTs from eligible persons’
- line 106: the authors should list out the brand(s) of Ag-RDTs used. Ag-RDTs divide into two versions, one is self-test version while the other one is professional version. Self-test version means that the whole procedures might be performed by laymen after going through the instruction manual. Professional version means that the whole procedures should be performed by healthcare provider or trained persons.
The authors are suggested to go through the following two sites:
1. WHO website for IVDs
https://extranet.who.int/pqweb/news/who-issues-its-first-emergency-use-listing-sars-cov-2-self-test
2. List of kits recommended by WHO for emergency use
https://extranet.who.int/pqweb/sites/default/files/documents/230207_EUL_SARS-CoV-2_Approved_IVDs.pdf
- one of the Eligibility criteria to enroll in the study was: within 7 days of onset of symptoms (lines 119-120, Figure 1), however, the results shown in Table 2 that 126 participants having symptoms more than days were also recruited. The authors should justify the discordant information presented between methods section and results section.
- line 244: the authors should elaborate more for the ‘non-random sample’ for those 239 participants performed for both PCR and RDT tests.
- discussion section: the authors should discuss the performance of Ag-RDTs varied between different brands. Policy makers should pay cautions in selecting the suitable Ag-RDTs for testing.
Minor comments:
- Extensive editing of English language required.
Example
Lines 43-45: Original: Therefore, it is likely that the number of people infected and deaths due to COVID-19 were higher than reported.
Suggested: Therefore, it is likely that the number of people infected and deaths due to COVID-19 were underestimated.
- The authors should use abbreviations in a systemically manner. If an abbreviation has appeared in the text, define it on first use in both places. After you define an abbreviation, use only the abbreviation.
Example
Lines 59 and 69: The term ‘Ag-RDTs’ was first mentioned in line 59, however, the authors presented the spelled-out version in line 69.
see my comments to authors mentioned above.
Reviewer 3 Report
The authors have investigated the feasibility, uptake, and results of Ag-RDTs and identified facilitators and barriers to Ag-RDT testing. In this study, the authors have enrolled a total of 27,888 participants, of whom 24,956 (89.5%) consented to testing and 121 (0.5%) were found positive. The authors demonstrated great feasibility in utilising Ag-RDTs as a screening and diagnostic tool for COVID-19 infections, with nearly 90% uptake. Authors have investigated the high specificity and negative predictive values and the higher positivity rates among severe COVID-19 symptomatic, so embedding Ag-RDTs into COVID-19 testing strategies for ruling out and in COVID-19 infections would hold a great advantage. The paper is generally well written and structured; my suggestions are as follows:
- Authors can mention inclusion and exclusion criteria for the enrollment of individuals.
- Authors should mention the limitations of their study.
here is a need for minor improvements in typos, language, and grammar control throughout the manuscript. For example, in the abstract
Background: Northwest Syria (NWS) is a conflict-affected and unstable area. Due to its limited health infrastructure, accessing advanced COVID-19 testing services is challenging. COVID-19 antigen rapid diagnostic tests (Ag-RDTs) have the potential to overcome this barrier. Therefore, a pilot project was implemented to introduce Ag-RDTs in this setting, aiming to a) describe the feasibility, uptake, and results of Ag-RDTs; and b) identify facilitators and barriers to Ag-RDT testing.
Methods: A cross-sectional study design involving secondary analysis of data collected during the project’s monitoring was developed. A local NGO implemented 25,000 Ag-RDTs across borders through trained community health workers.
Results: A total of 27,888 persons were found eligible and enrolled, of whom 24,956 (89.5%) consented to test, and 121 (0.5%) were found positive. Highest positivity was observed among those with severe COVID-19 symptoms (12.7%), those with respiratory illnesses (2.5%), persons enrolled at Afrin Hospitals (2.5%), and healthcare workers (1.9%). A non-random sample of 236 people underwent a confirmatory RT-PCR test. Accordingly, the observed sensitivity, specificity, positive, and negative predictive values were 80.0%, 96.1%, 91.4%, and 90.3%, respectively. Key challenges encountered included obtaining informed consent and conducting confirmatory RT-PCR testing. 4)
Conclusion: This project demonstrated great feasibility in utilising Ag-RDTs as a screening and diagnostic tool for COVID-19 infections, with nearly 90% uptake. Considering the high specificity and negative predictive values, and the higher positivity rates among severe COVID-19 symptomatic, embedding Ag-RDTs into COVID-19 testing strategies for ruling out and in COVID-19 infections would hold a great advantage.
Round 2
Reviewer 1 Report
The comments have been addressed. The reported diagnostic properties for the RDT within text between lines 128 and 132 do not match with the results presented in table where PCR was used as gold standard. This should be addressed. Since the diagnostic properties of the PCR is unknown and the sampling for the PCR testing was non-randomized the text between lines 373 and 382 is misleading and should be removed.
Author Response
Author’s response:
Many thanks for your review and suggestions. We have tried our best to address your concerns/suggestions.
The reported diagnostic properties in lines 128-132 is that of the manufacturers and we have indicated that as "manufacturer reported” properties (revised manuscript line numbers 112-113).
Though the diagnostic properties of the RT-PCR tests used at various laboratories in NWS were unknown, for all practical purposes, the RT-PCR tests can be considered as the standard of care test in the local context which the Ag-RDT is expected to emulate. Therefore, we have compared the performance of the Ag-RDTs with respect to the available RT-PCR tests and have called them ‘observed’ diagnostic properties. (Revised manuscript line numbers 313-314).
We hope the use of the terms ‘manufacturer reported’ and ‘observed’ address the concern of the reviewer.
As suggested, we have deleted the text on the diagnostic properties in the discussion section (revised manuscript line numbers 391-394)

Reviewer 2 Report
The authors addressed all of my queries.
The authors said that they used the English editing service.
Author Response
Dear Reviewer,
Many thanks for reviewing and accepting our responses.
Warm regards,
Hassan Ghawji.